# Glyce: Glyph-vectors for Chinese Character Representations

**Yuxian Meng\*, Wei Wu\*, Fei Wang\*, Xiaoya Li\*, Ping Nie, Fan Yin**
**Muyu Li, Qinghong Han, Xiaofei Sun and Jiwei Li**

Shannon.AI
{yuxian_meng, wei_wu, fei_wang, xiaoya_li, ping_nie, fan_yin,
muyu_li, qinghong_han, xiaofei_sun, jiwei_li}@shannonai.com

## Abstract

It is intuitive that NLP tasks for logographic languages like Chinese should benefit from the use of the glyph information in those languages. However, due to the lack of rich pictographic evidence in glyphs and the weak generalization ability of standard computer vision models on character data, an effective way to utilize the glyph information remains to be found.

In this paper, we address this gap by presenting Glyce, the glyph-vectors for Chinese character representations. We make three major innovations: (1) We use historical Chinese scripts (e.g., bronzeware script, seal script, traditional Chinese, etc) to enrich the pictographic evidence in characters; (2) We design CNN structures (called tianzege-CNN) tailored to Chinese character image processing; and (3) We use image-classification as an auxiliary task in a multi-task learning setup to increase the model's ability to generalize.

We show that glyph-based models are able to consistently outperform word/char ID-based models in a wide range of Chinese NLP tasks. We are able to set new state-of-the-art results for a variety of Chinese NLP tasks, including tagging (NER, CWS, POS), sentence pair classification, single sentence classification tasks, dependency parsing, and semantic role labeling. For example, the proposed model achieves an F1 score of 80.6 on the OntoNotes dataset of NER, +1.5 over BERT; it achieves an almost perfect accuracy of 99.8% on the Fudan corpus for text classification. [1] [2]

## 1   Introduction

Chinese is a logographic language. The logograms of Chinese characters encode rich information of their meanings. Therefore, it is intuitive that NLP tasks for Chinese should benefit from the use of the glyph information. Taking into account logographic information should help semantic modeling. Recent studies indirectly support this argument: Radical representations have proved to be useful in a wide range of language understanding tasks [Shi et al., 2015, Li et al., 2015, Yin et al., 2016, Sun et al., 2014, Shao et al., 2017]. Using the Wubi scheme — a Chinese character encoding method that mimics the order of typing the sequence of radicals for a character on the computer keyboard —- is reported to improve performances on Chinese-English machine translation [Tan et al., 2018]. Cao et al. [2018] gets down to units of greater granularity, and proposed stroke n-grams for character modeling.

Recently, there have been some efforts applying CNN-based algorithms on the visual features of characters. Unfortunately, they do not show consistent performance boosts [Liu et al., 2017, Zhang

| Chinese | English | Time Period |
|---|---|---|
| 金文 | Bronzeware script | Shang and Zhou dynasty (2000 BC – 300 BC) |
| 隶书 | Clerical script | Han dynasty (200BC-200AD) |
| 篆书 | Seal script | Han dynasty and Wei-Jin period (100BC - 420 AD) |
| 魏碑 | Tablet script | Northern and Southern dynasties 420AD - 588AD |
| 繁体中文 | Traditional Chinese | 600AD - 1950AD (mainland China). still currently used in HongKong and Taiwan |
| 简体中文(宋体) | Simplified Chinese - Song | 1950-now |
| 简体中文(仿宋体) | Simplified Chinese - FangSong | 1950-now |
| 草书 | Cursive script | Jin Dynasty to now |

Table 1: Scripts and writing styles used in Glyce.

and LeCun, 2017], and some even yield negative results [Dai and Cai, 2017]. For instance, Dai and Cai [2017] run CNNs on char logos to obtain Chinese character representations and used them in the downstream language modeling task. They reported that the incorporation of glyph representations actually worsens the performance and concluded that CNN-based representations do not provide extra useful information for language modeling. Using similar strategies, Liu et al. [2017] and Zhang and LeCun [2017] tested the idea on text classification tasks, and performance boosts were observed only in very limited number of settings. Positive results come from Su and Lee [2017], which found glyph embeddings help two tasks: word analogy and word similarity. Unfortunately, Su and Lee [2017] only focus on word-level semantic tasks and do not extend improvements in the word-level tasks to higher level NLP tasks such as phrase, sentence or discourse level. Combined with radical representations, Shao et al. [2017] run CNNs on character figures and use the output as auxiliary features in the POS tagging task.

We propose the following explanations for negative results reported in the earlier CNN-based models [Dai and Cai, 2017]: (1) not using the correct version(s) of scripts: Chinese character system has a long history of evolution. The characters started from being easy-to-draw, and slowly transitioned to being easy-to-write. Also, they became less pictographic and less concrete over time. The most widely used script version to date, the *Simplified Chinese*, is the easiest script to write, but inevitably loses the most significant amount of pictographic information. For example, "人" (human) and "入" (enter), which are of irrelevant meanings, are highly similar in shape in simplified Chinese, but very different in historical languages such as bronzeware script. (2) not using the proper CNN structures: unlike ImageNet images [Deng et al., 2009], the size of which is mostly at the scale of 800*600, character logos are significantly smaller (usually with the size of 12*12). It requires a different CNN architecture to capture the local graphic features of character images; (3) no regulatory functions were used in previous work: unlike the classification task on the imageNet dataset, which contains tens of millions of data points, there are only about 10,000 Chinese characters. Auxiliary training objectives are thus critical in preventing overfitting and promoting the model's ability to generalize.

In this paper, we propose GLYCE, the GLYph-vectors for Chinese character representations. We treat Chinese characters as images and use CNNs to obtain their representations. We resolve the aforementioned issues by using the following key techniques:

- We use the ensemble of the historical and the contemporary scripts (e.g., the bronzeware script, the clerical script, the seal script, the traditional Chinese etc), along with the scripts of different writing styles (e.g, the cursive script) to enrich pictographic information from the character images.
- We utilize the Tianzige-CNN (田字格) structures tailored to logographic character modeling.
- We use multi-task learning methods by adding an image-classification loss function to increase the model's ability to generalize.

Glyce is found to improve a wide range of Chinese NLP tasks. We are able to obtain the SOTA performances on a wide range of Chinese NLP tasks, including tagging (NER, CWS, POS), sentence pair classification (BQ, LCQMC, XNLI, NLPCC-DBQA), single sentence classification tasks (ChnSentiCorp, the Fudan corpus, iFeng), dependency parsing, and semantic role labeling.

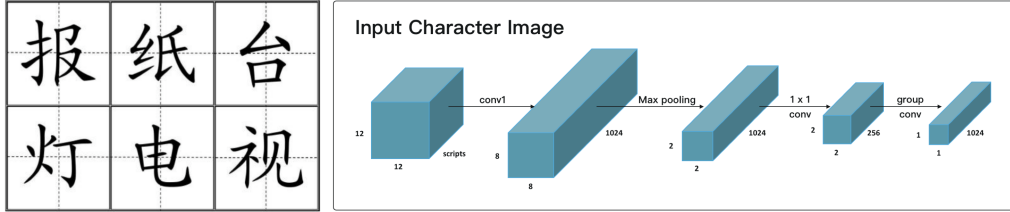

Figure 1: Illustration of the Tianzege-CNN used in Glyce.

## 2 Glyce

### 2.1 Using Historical Scripts

As discussed in Section 1, pictographic information is heavily lost in the simplified Chinese script. We thus propose using scripts from various time periods in history and also of different writing styles. We collect the following major historical script with details shown in Table 1. Scripts from different historical periods, which are usually very different in shape, help the model to integrate pictographic evidence from various sources; Scripts of different writing styles help improve the model's ability to generalize. Both strategies are akin to widely-used data augmentation strategies in computer vision.

### 2.2 The Tianzige-CNN Structure for Glyce

Directly using deep CNNs He et al. [2016], Szegedy et al. [2016], Ma et al. [2018a] in our task results in very poor performances because of (1) relatively smaller size of the character images: the size of Imagenet images is usually at the scale of 800*600, while the size of Chinese character images is significantly smaller, usually at the scale of 12*12; and (2) the lack of training examples: classifications on the imageNet dataset utilizes tens of millions of different images. In contrast, there are only about 10,000 distinct Chinese characters. To tackle these issues, we propose the Tianzige-CNN structure, which is tailored to Chinese character modeling as illustrated in Figure 1. Tianzige (田字格) is a traditional form of Chinese Calligraphy. It is a four-squared format (similar to Chinese character 田) for beginner to learn writing Chinese characters. The input image $x_{\text{image}}$ is first passed through a convolution layer with kernel size 5 and output channels 1024 to capture lower level graphic features. Then a max-pooling of kernel size 4 is applied to the feature map which reduces the resolution from $8 \times 8$ to $2 \times 2$, . This $2 \times 2$ tianzige structure presents how radicals are arranged in Chinese characters and also the order by which Chinese characters are written. Finally, we apply group convolutions [Krizhevsky et al., 2012, Zhang et al., 2017] rather than conventional convolutional operations to map tianzige grids to the final outputs . Group convolutional filters are much smaller than their normal counterparts, and thus are less prone to overfitting. It is fairly easy to adjust the model from single script to multiple scripts, which can be achieved by simply changing the input from 2D (i.e., $d_{\text{font}} \times d_{\text{font}}$) to 3D (i.e., $d_{\text{font}} \times d_{\text{font}} \times N_{\text{script}}$), where $d_{\text{font}}$ denotes the font size and $N_{\text{script}}$ the number of scripts we use.

### 2.3 Image Classification as an Auxiliary Objective

To further prevent overfitting, we use the task of image classification as an auxiliary training objective. The glyph embedding $h_{\text{image}}$ from CNNs will be forwarded to an image classification objective to predict its corresponding charID. Suppose the label of image $x$ is $z$. The training objective for the image classification task $\mathcal{L}(\text{cls})$ is given as follows:

$$\begin{aligned} \mathcal{L}(\text{cls}) &= -\log p(z|x) \\ &= -\log \text{softmax}(W \times h_{\text{image}}) \end{aligned} \quad (1)$$

Let $\mathcal{L}(\text{task})$ denote the task-specific objective for the task we need to tackle, e.g., language modeling, word segmentation, etc. We linearly combine $\mathcal{L}(\text{task})$ and $\mathcal{L}(\text{cl})$, making the final objective training function as follows:

$$\mathcal{L} = (1 - \lambda(t))\,\mathcal{L}(\text{task}) + \lambda(t)\mathcal{L}(\text{cls}) \quad (2)$$

where $\lambda(t)$ controls the trade-off between the task-specific objective and the auxiliary image-classification objective. $\lambda$ is a function of the number of epochs $t$: $\lambda(t) = \lambda_0 \lambda_1^t$, where $\lambda_0 \in [0, 1]$

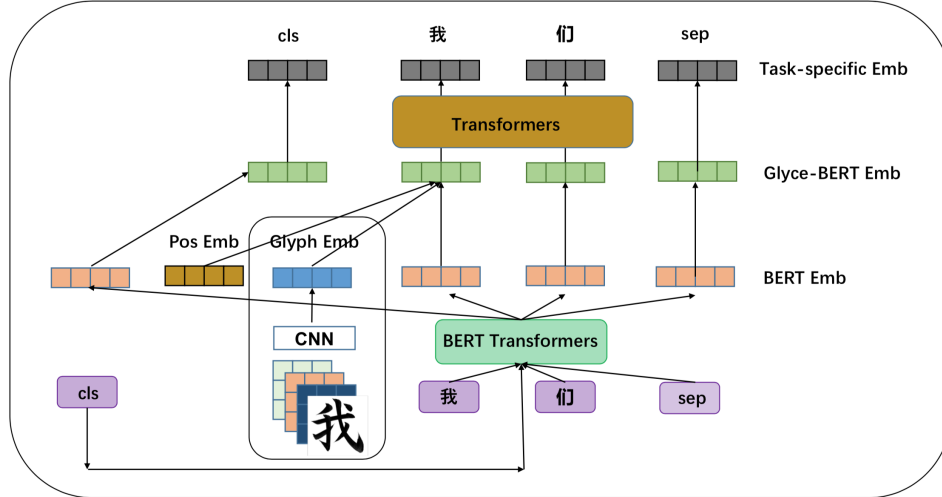

Figure 2: Combing glyph information with BERT.

denotes the starting value, $\lambda_1 \in [0, 1]$ denotes the decaying value. This means that the influence from the image classification objective decreases as the training proceeds, with the intuitive explanation being that at the early stage of training, we need more regulations from the image classification task. Adding image classification as a training objective mimics the idea of multi-task learning.

## 2.4 Combing Glyph Information with BERT

The glyph embeddings can be directly output to downstream models such as RNNs, LSTMs, transformers.

Since large scale pretraining systems using language models, such as BERT [Devlin et al., 2018], ELMO [Peters et al., 2018] and GPT [Radford et al., 2018], have proved to be effective in a wide range of NLP tasks, we explore the possibility of combining glyph embeddings with BERT embeddings. Such a strategy will potentially endow the model with the advantage of both glyph evidence and large-scale pretraining. The overview of the combination is shown in Figure 2. The model consists of four layers: the BERT layer, the glyph layer, the Glyce-BERT layer and the task-specific output layer.

- **BERT Layer** Each input sentence $S$ is concatenated with a special CLS token denoting the start of the sentence, and a SEP token, denoting the end of the sentence. Given a pre-trained BERT model, the embedding for each token of $S$ is computed using BERT. We use the output from the last layer of the BERT transformer to represent the current token.

- **Glyph Layer** the output glyph embeddings of $S$ from tianzege-CNNs.

- **Glyce-BERT layer** Position embeddings are first added to the glyph embeddings. The addition is then concatenated with BERT to obtain the full Glyce representations.

- **Task-specific output layer** Glyce representations are used to represent the token at that position, similar as word embeddings or Elmo emebddings [Peters et al., 2018]. Contextual-aware information has already been encoded in the BERT representation but not glyph representations. We thus need additional context models to encode contextual-aware glyph representations. Here, we choose multi-layer transformers [Vaswani et al., 2017]. The output representations from transformers are used as inputs to the prediction layer. It is worth noting that the representations the special CLS and SEP tokens are maintained at the final task-specific embedding layer.

## 3 Tasks

In this section, we describe how glypg embeddings can be used for different NLP tasks. In the vanilla version, glyph embeddings are simply treated as character embeddings, which are fed to models built on top of the word-embedding layers, such as RNNs, CNNs or more sophisticated ones. If combined

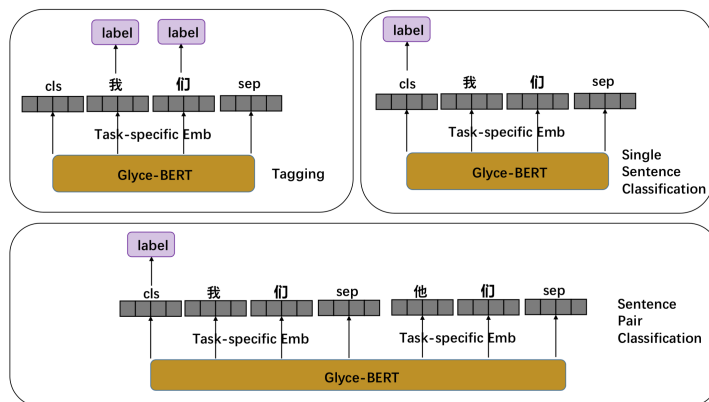

Figure 3: Using Glyce-BERT model for different tasks.

with BERT, we need to specifically handle the integration between the glyph embeddings and the pretrained embeddings from BERT in different scenarios, as will be discussed in order below:

**Sequence Labeling Tasks**    Many Chinese NLP tasks, such as name entity recognition (NER), Chinese word segmentation (CWS) and part speech tagging (POS), can be formalized as character-level sequence labeling tasks, in which we need to predict a label for each character. For glyce-BERT model, the embedding output from the task-specific layer (described in Section 2.4) is fed to the CRF model for label predictions.

**Single Sentence Classification**    For text classification tasks, a single label is to be predicted for the entire sentence. In the BERT model, the representation for the CLS token in the final layer of BERT is output to the softmax layer for prediction. We adopt the similar strategy, in which the representation for the CLS token in the task-specific layer is fed to the softmax layer to predict labels.

**Sentence Pair Classification**    For sentence pair classification task like SNIS [Bowman et al., 2015], a model needs to handle the interaction between the two sentences and outputs a label for a pair of sentences. In the BERT setting, a sentence pair $(s_1, s_2)$ is concatenated with one CLS and two SEP tokens, denoted by [CLS, $s_1$, SEP, $s_2$, SEP]. The concatenation is fed to the BERT model, and the obtained CLS representation is then fed to the softmax layer for label prediction. We adopt the similar strategy for Glyce-BERT, in which [CLS, $s_1$, SEP, $s_2$, SEP] is subsequently passed through the BERT layer, Glyph layer, Glyce-BERT layer and the task-specific output layer. The CLS representation from the task-specific output layer is fed to the softmax function for the final label prediction.

## 4    Experiments

To enable apples-to-apples comparison, we perform grid parameter search for both baselines and the proposed model on the dev set. Tasks that we work on are described in order below.

### 4.1    Tagging

**NER**    For the task of Chinese NER, we used the widely-used OntoNotes, MSRA, Weibo and resume datasets. Since most datasets don't have gold-standard segmentation, the task is normally treated as a char-level tagging task: outputting an NER tag for each character. The currently most widely used non-BERT model is Lattice-LSTMs [Yang et al., 2018, Zhang and Yang, 2018], achieving better performances than CRF+LSTM [Ma and Hovy, 2016].

**CWS**    : The task of Chinese word segmentation (CWS) is normally treated as a char-level tagging problem. We used the widely-used PKU, MSR, CITYU and AS benchmarks from SIGHAN 2005 bake-off for evaluation.

**POS** The task of Chinese part of speech tagging is normally formalized as a character-level sequence labeling task, assigning labels to each of the characters within the sequence. We use the CTB5, CTB9 and UD1 (Universal Dependencies) benchmarks to test our models.

| OntoNotes | | | | Weibo | | | |
|---|---|---|---|---|---|---|---|
| Model | P | R | F | Model | P | R | F |
| CRF-LSTM | 74.36 | 69.43 | 71.81 | CRF-LSTM | 51.16 | 51.07 | 50.95 |
| Lattice-LSTM | 76.35 | 71.56 | 73.88 | Lattice-LSTM | 52.71 | 53.92 | 53.13 |
| Glyce+Lattice-LSTM | 82.06 | 68.74 | 74.81 | Lattice-LSTM+Glyce | 53.69 | 55.30 | 54.32 |
| | | | **(+ 0.93)** | | | | **(+1.19)** |
| BERT | 78.01 | 80.35 | 79.16 | BERT | 67.12 | 66.88 | 67.33 |
| Glyce+BERT | 81.87 | 81.40 | 80.62 | Glyce+BERT | 67.68 | 67.71 | 67.60 |
| | | | **(+1.46)** | | | | **(+0.27)** |
| MSRA | | | | resume | | | |
| Model | P | R | F | Model | P | R | F |
| CRF-LSTM | 92.97 | 90.80 | 91.87 | CRF-LSTM | 94.53 | 94.29 | 94.41 |
| Lattice-LSTM | 93.57 | 92.79 | 93.18 | Lattice-LSTM | 94.81 | 94.11 | 94.46 |
| Lattice-LSTM+Glyce | 93.86 | 93.92 | 93.89 | Lattice-LSTM+Glyce | 95.72 | 95.63 | 95.67 |
| | | | **(+0.71)** | | | | **(+1.21)** |
| BERT | 94.97 | 94.62 | 94.80 | BERT | 96.12 | 95.45 | 95.78 |
| Glyce+BERT | 95.57 | 95.51 | 95.54 | Glyce+BERT | 96.62 | 96.48 | 96.54 |
| | | | **(+0.74)** | | | | **(+0.76)** |

Table 2: Results for NER tasks.

Results for NER, CWS and POS are respectively shown in Tables 2, 3 and 4. When comparing with non-BERT models, Lattice-Glyce performs better than all non-BERT models across all datasets in all tasks. BERT outperforms non-BERT models in all datasets except Weibo. This is due to the discrepancy between the dataset which BERT is pretrained on (i.e., wikipedia) and weibo. The Glyce-BERT model outperforms BERT and sets new SOTA results across all datasets, manifesting the effectiveness of incorporating glyph information. We are able to achieve SOTA performances on all of the datasets using either Glyce model itself or BERT-Glyce model.

| PKU | | | | CITYU | | | |
|---|---|---|---|---|---|---|---|
| Model | P | R | F | Model | P | R | F |
| Yang et al. [2017] | - | - | 96.3 | Yang et al. [2017] | - | - | 96.9 |
| Ma et al. [2018b] | - | - | 96.1 | Ma et al. [2018b] | - | - | 97.2 |
| Huang et al. [2019] | - | - | 96.6 | Huang et al. [2019] | - | - | 97.6 |
| BERT | 96.8 | 96.3 | 96.5 | BERT | 97.5 | 97.7 | 97.6 |
| Glyce+BERT | 97.1 | 96.4 | 96.7 | Glyce+BERT | 97.9 | 98.0 | 97.9 |
| | | | **(+0.2)** | | | | **(+0.3)** |
| MSR | | | | AS | | | |
| Model | P | R | F | Model | P | R | F |
| Yang et al. [2017] | - | - | 97.5 | Yang et al. [2017] | - | - | 95.7 |
| Ma et al. [2018b] | - | - | 98.1 | Ma et al. [2018b] | - | - | 96.2 |
| Huang et al. [2019] | - | - | 97.9 | Huang et al. [2019] | - | - | 96.6 |
| BERT | 98.1 | 98.2 | 98.1 | BERT | 96.7 | 96.4 | 96.5 |
| Glyce+BERT | 98.2 | 98.3 | 98.3 | Glyce+BERT | 96.6 | 96.8 | 96.7 |
| | | | **(+0.2)** | | | | **(+0.2)** |

Table 3: Results for CWS tasks.

## 4.2 Sentence Pair Classification

For sentence pair classification tasks, we need to output a label for each pair of sentence. We employ the following four different datasets: (1) **BQ** (binary classification task) [Bowman et al., 2015]; (2) **LCQMC** (binary classification task) [Liu et al., 2018], (3) **XNLI** (three-class classification task) [Williams and Bowman], and (4) **NLPCC-DBQA** (binary classification task) [3].

## Table 4

### CTB5

| Model | P | R | F |
|---|---|---|---|
| Shao et al. [2017] (Sig) | 93.68 | 94.47 | 94.07 |
| Shao et al. [2017] (Ens) | 93.95 | 94.81 | 94.38 |
| Lattice-LSTM | 94.77 | 95.51 | 95.14 |
| Glyce+Lattice-LSTM | 95.49 | 95.72 | 95.61 |
| | | | **(+0.47)** |
| BERT | 95.86 | 96.26 | 96.06 |
| Glyce+BERT | 96.50 | 96.74 | 96.61 |
| | | | **(+0.55)** |

### CTB9

| Model | P | R | F |
|---|---|---|---|
| Shao et al. [2017] (Sig) | 91.81 | 94.47 | 91.89 |
| Shao et al. [2017] (Ens) | 92.28 | 92.40 | 92.34 |
| Lattice-LSTM | 92.53 | 91.73 | 92.13 |
| Lattice-LSTM+Glyce | 92.28 | 92.85 | 92.38 |
| | | | **(+0.25)** |
| BERT | 92.43 | 92.15 | 92.29 |
| Glyce+BERT | 93.49 | 92.84 | 93.15 |
| | | | **(+0.86)** |

### CTB6

| Model | P | R | F |
|---|---|---|---|
| Shao et al. [2017] (Sig) | - | - | 90.81 |
| Lattice-LSTM | 92.00 | 90.86 | 91.43 |
| Glyce+Lattice-LSTM | 92.72 | 91.14 | 91.92 |
| | | | **(+0.49)** |
| BERT | 94.91 | 94.63 | 94.77 |
| Glyce+BERT | 95.56 | 95.26 | 95.41 |
| | | | **(+0.64)** |

### UD1

| Model | P | R | F |
|---|---|---|---|
| Shao et al. [2017] (Sig) | 89.28 | 89.54 | 89.41 |
| Shao et al. [2017] (Ens) | 89.67 | 89.86 | 89.75 |
| Lattice-LSTM | 90.47 | 89.70 | 90.09 |
| Lattice-LSTM+Glyce | 91.57 | 90.19 | 90.87 |
| | | | **(+0.78)** |
| BERT | 95.42 | 94.17 | 94.79 |
| Glyce+BERT | 96.19 | 96.10 | 96.14 |
| | | | **(+1.35)** |

Table 4: Results for POS tasks.

The current non-BERT SOTA model is based on the bilateral multi-perspective matching model (BiMPM) [Wang et al., 2017], which specifically tackles the subunit matching between sentences. Glyph embeddings are incorporated into BiMPMs, forming the Glyce+BiMPM baseline. Results regarding each model on different datasets are given in Table 5. As can be seen, BiPMP+Glyce outperforms BiPMPs, achieving the best results among non-bert models. BERT outperforms all non-BERT models, and BERT+Glyce performs the best, setting new SOTA results on all of the four benchmarks.

## Table 5

### BQ

| Model | P | R | F | A |
|---|---|---|---|---|
| BiMPM | 82.3 | 81.2 | 81.7 | 81.9 |
| Glyce+BiMPM | 81.9 | 85.5 | 83.7 | 83.3 |
| | | | **(+2.0)** | **(+1.4)** |
| BERT | 83.5 | 85.7 | 84.6 | 84.8 |
| Glyce+BERT | 84.2 | 86.9 | 85.5 | 85.8 |
| | | | **(+0.9)** | **(+1.0)** |

### LCQMC

| Model | P | R | F | A |
|---|---|---|---|---|
| BiMPM | 77.6 | 93.9 | 85.0 | 83.4 |
| Glyce+BiMPM | 80.4 | 93.4 | 86.4 | 85.3 |
| | | | **(+1.4)** | **(+1.9)** |
| BERT | 83.2 | 94.2 | 88.2 | 87.5 |
| Glyce+BERT | 86.8 | 91.2 | 88.8 | 88.7 |
| | | | **(+0.6)** | **(+1.2)** |

### XNLI

| Model | P | R | F | A |
|---|---|---|---|---|
| BiMPM | - | - | - | 67.5 |
| Glyce+BiMPM | - | - | - | 67.7 |
| | | | | **(+0.2)** |
| BERT | - | - | - | 78.4 |
| Glyce+BERT | - | - | - | 79.2 |
| | | | | **(+0.8)** |

### NLPCC-DBQA

| Model | P | R | F | A |
|---|---|---|---|---|
| BiMPM | 78.8 | 56.5 | 65.8 | - |
| Glyce+BiMPM | 76.3 | 59.9 | 67.1 | - |
| | | | **(+1.3)** | - |
| BERT | 79.6 | 86.0 | 82.7 | - |
| Glyce+BERT | 81.1 | 85.8 | 83.4 | - |
| | | | **(+0.7)** | - |

Table 5: Results for sentence-pair classification tasks.

## Table 6

| Model | ChnSentiCorp | the Fudan corpus | iFeng |
|---|---|---|---|
| LSTM | 91.7 | 95.8 | 84.9 |
| LSTM + Glyce | 93.1 | 96.3 | 85.8 |
| | **(+ 1.4)** | **(+0.5)** | **(+0.9)** |
| BERT | 95.4 | 99.5 | 87.1 |
| Glyce+BERT | 95.9 | 99.8 | 87.5 |
| | **(+0.5)** | **(+0.3)** | **(+0.4)** |

Table 6: Accuracies for Single Sentence Classification task.

| Dependency Parsing | | |
|---|---|---|
| Model | UAS | LAS |
| Ballesteros et al. [2016] | 87.7 | 86.2 |
| Kiperwasser and Goldberg [2016] | 87.6 | 86.1 |
| Cheng et al. [2016] | 88.1 | 85.7 |
| Biaffine | 89.3 | 88.2 |
| Biaffine+Glyce | 90.2 **(+0.9)** | 89.0 **(+0.8)** |

| Semantic Role Labeling | | | |
|---|---|---|---|
| Model | P | R | F |
| Roth and Lapata [2016] | 76.9 | 73.8 | 75.3 |
| Marcheggiani and Titov [2017] | 84.6 | 80.4 | 82.5 |
| He et al. [2018] | 84.2 | 81.5 | 82.8 |
| k-order pruning+Glyce | 85.4 **(+0.8)** | 82.1 **(+0.6)** | 83.7 **(+0.9)** |

Table 7: Results for dependency parsing and SRL.

### 4.3 Single Sentence Classification

For single sentence/document classification, we need to output a label for a text sequence. The label could be either a sentiment indicator or a news genre. Datasets that we use include: (1) ChnSentiCorp (binary classification); (2) the Fudan corpus (5-class classification) [Li, 2011]; and (3) Ifeng (5-class classification).

Results for different models on different tasks are shown in Table 6. We observe similar phenomenon as before: Glyce+BERT achieves SOTA results on all of the datasets. Specifically, the Glyce+BERT model achieves an almost perfect accuracy (99.8) on the Fudan corpus.

### 4.4 Dependency Parsing and Semantic Role Labeling

For dependency parsing [Chen and Manning, 2014, Dyer et al., 2015], we used the widely-used Chinese Penn Treebank 5.1 dataset for evaluation. Our implementation uses the previous state-of-the-art Deep Biaffine model Dozat and Manning [2016] as a backbone. We replaced the word vectors from the biaffine model with Glyce-word embeddings, and exactly followed its model structure and training/dev/test split criteria. We report scores for unlabeled attachment score (UAS) and labeled attachment score (LAS). Results for previous models are copied from [Dozat and Manning, 2016, Ballesteros et al., 2016, Cheng et al., 2016]. Glyce-word pushes SOTA performances up by +0.9 and +0.8 in terms of UAS and LAS scores.

For the task of semantic role labeling (SRL) [Roth and Lapata, 2016, Marcheggiani and Titov, 2017, He et al., 2018], we used the CoNLL-2009 shared-task. We used the current SOTA model, the k-order pruning algorithm [He et al., 2018] as a backbone.[4] We replaced word embeddings with Glyce embeddings. Glyce outperforms the previous SOTA performance by 0.9 with respect to the F1 score, achieving a new SOTA score of 83.7.

BERT does not perform competitively in these two tasks, and results are thus omitted.

## 5 Ablation Studies

In this section, we discuss the influence of different factors of the proposed model. We use the LCQMC dataset of the sentence-pair prediction task for illustration. Factors that we discuss include training strategy, model architecture, auxiliary image-classification objective, etc.

### 5.1 Training Strategy

This section talks about a training tactic (denoted by BERT-glyce-joint), in which given task-specific supervisions, we first fine-tune the BERT model, then freeze BERT to fine-tune the glyph layer, and finally jointly tune both layers until convergence. We compare this strategy with other tactics, including (1) the *Glyph-Joint* strategy, in which BERT is not fine-tuned in the beginning: we first

| Strategy | P | R | F | Acc |
|---|---|---|---|---|
| BERT-glyce-joint | 86.8 | 91.2 | **88.8** | **88.7** |
| Glyph-Joint | 82.5 | 94.0 | 87.9 | 87.1 |
| joint | 81.5 | 95.1 | 87.8 | 86.8 |
| only BERT | 83.2 | 94.2 | 88.2 | 87.5 |

Table 8: Impact of different training strategies.

| Strategy | Precision | Recall | F1 | Accuracy |
|---|---|---|---|---|
| Transformers | 86.8 | 91.2 | **88.8** | **88.7** |
| BiLSMTs | 81.8 | 94.9 | 87.9 | 86.9 |
| CNNs | 81.5 | 94.8 | 87.6 | 86.6 |
| BiMPM | 81.1 | 94.6 | 87.3 | 86.2 |

Table 9: Impact of structures for the task-specific output layer.

| Strategy | P | R | F | Acc |
|---|---|---|---|---|
| W image-cls | 86.8 | 91.2 | **88.8** | **88.7** |
| WO image-cls | 83.9 | 93.6 | 88.4 | 87.9 |

Table 10: Impact of the auxilliary image classification training objective.

| | P | R | F |
|---|---|---|---|
| Vanilla-CNN | 85.3 | 89.8 | 87.4 |
| He et al. [2016] | 84.5 | 90.8 | 87.5 |
| Tianzige-CNN | 86.8 | 91.2 | **88.8** |

Table 11: Impact of CNN structures.

freeze BERT to tune the glyph layer, and then jointly tune both layers until convergence; and (2) the *joint* strategy, in which we directly jointly training two models until converge.

Results are shown in Table 8. As can be seen, the BERT-glyce-joint outperforms the rest two strategies. Our explanation for the inferior performance of the *joint* strategy is as follows: the BERT layer is pretrained but the glyph layer is randomly initialized. Given the relatively small amount of training signals, the BERT layer could be mislead by the randomly initialized glyph layer at the early stage of training, leading to inferior final performances.

## 5.2 Structures of the task-specific output layer

The concatenation of the glyph embedding and the BERT embedding is fed to the task-specific output layer. The task-specific output layer is made up with two layers of transformer layers. Here we change transformers to other structures such as BiLSTMs and CNNs to explore the influence. We also try the BiMPM structure Wang et al. [2017] to see the results.

Performances are shown in Table 9. As can be seen, transformers not only outperform BiLSTMs and CNNs, but also the BiMPM structure, which is specifically built for the sentence pair classification task. We conjecture that this is because of the consistency between transformers and the BERT structure.

## 5.3 The image-classification training objective

We also explore the influence of the image-classification training objective, which outputs the glyph representation to an image-classification objective. Table 10 represents its influence. As can be seen, this auxiliary training objective given a +0.8 performance boost.

## 5.4 CNN structures

Results for different CNN structures are shown in Table 11. As can be seen, the adoption of tianzige-CNN structure introduces a performance boost of F1 about +1.0. Directly using deep CNNs in our task results in very poor performances because of (1) relatively smaller size of the character images: the size of ImageNet images is usually at the scale of 800*600, while the size of Chinese character images is significantly smaller, usually at the scale of 12*12; and (2) the lack of training examples: classifications on the ImageNet dataset utilizes tens of millions of different images. In contrast, there are only about 10,000 distinct Chinese characters. We utilize the Tianzige-CNN (田字格) structures tailored to logographic character modeling for Chinese. This tianzige structure is of significant importance in extracting character meanings.

## 6 Conclusion

In this paper, we propose Glyce, Glyph-vectors for Chinese Character Representations. Glyce treats Chinese characters as images and uses Tianzige-CNN to extract character semantics. Glyce provides a general way to model character semantics of logographic languages. It is general and fundamental. Just like word embeddings, Glyce can be integrated to any existing deep learning system.

## Footnotes

[1] \* indicates equal contribution.

[2] code is available at `https://github.com/ShannonAI/glyce`.

[3] https://github.com/xxx0624/QA_Model

[4]Code open sourced at `https://github.com/bcmi220/srl_syn_pruning`

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
