[Reviews · NeurIPS 2019]

Reviewer 1



This paper describes a method for leveraging sub-character information from Chinese characters, and reports small but reliable improvements on a large number of Chinese NLP tasks. The paper is strong in the results that it reports. The authors show that incorporation of their "Glyce" embeddings improves results from BERT (which is SOTA on nearly all of the tasks), as well the strongest non-BERT models, for a wide variety of tasks. So it appears that the authors' methods have successfully allowed them to leverage some useful signal from the sub-character information, which seems a reasonably significant contribution for Chinese NLP. The main weakness of the paper is in clarity of the methods. I'm not clear on the details of the training procedure. Is the glyph CNN trained jointly with each separate downstream task? Or is the CNN pre-trained separately such that it can be used ELMo-style on a variety of downstream tasks, as certain sections seem to suggest? If the latter is the case, then I'm not clear on the nature of the "task" objective that combines with the image classification objective. I would also like to see more detail about the glyph CNN itself. Is the image classification predicting the character identities (10k possible labels?). Does the model receive at each timestep the relevant character in all of the mentioned scripts? (If not, how are the different scripts being leveraged?) Finally, there should be more detail about the tasks. Little information is given about the sentence pair and single-sentence classification tasks in particular, giving mostly just the acronym designations and number of classes, rather than detail about the task itself. Relatedly, I would like to see some discussion of why we would expect the character information to help with these different tasks. An additional point: currently it is difficult to discern which of the authors' innovations contribute most to improving performance -- ablation experiments would be helpful in this regard. Minor: Text in Figure 2 is too small to read. Sec 4.1 "BERT outperforms non-BERT models in all datasets except Weibo" -- is this true? BERT seems to be better on Weibo as well. line 95: prune --> prone line 124: use to --> use the line 174: character --> characters line 201: the the

Reviewer 2



Regarding the first contribution: using visual features for Chinese characters (which are visually inspired) is a long-standing idea, but as the authors pointed out, few previous works were able to achieve a significantly better performance than purely embedding driven approaches. In this regard, the paper does a good revisit to the problem and provides an interesting hypothesis. Regarding the second contribution: a dedicated CNN structure with diverse training corpus (historical scripts) and multi-task learning scenario is novel and seems to be effective, as demonstrated through several benchmark datasets. It is especially encouraging to see that this gives non-trivial additional improvement over BERT, a very strong baseline. Regarding the lack of analysis: this is a rather disappointing because this kind of paper would benefit from a qualtitative evidence that backs how the visual features help the downstream task-specific model to better understand the language. Appropriateness: I am also worried about the appropriateness of the paper to NeurIPS. The paper seems to be more appropriate for NLP conferences than NeurIPS. I am not sure if NeurIPS has a large-enough audience for this kind of work. After rebuttal: the rebuttal was helpful but more detailed analysis on why/how a visual model helps to learn better character embedding would be desired. Increased my score from 5 to 6.

Reviewer 3



The authors propose a very simple approach to capturing the pictographic nature of Chinese characters into embeddings. The goal in doing so being that such a representation captures semantic similarities perhaps obvious to the eye but difficult to learn from clustering the character IDs in a downstream task. Further, the authors provide insights into why previous approaches failed to see performance gains.

[Author Response · NeurIPS 2019]

1 We sincerely thank all the reviewers for their insightful suggestions.

2 **1 Ablation Studies** The issues raised by the reviewers on ablation studies are very sensible. Actually, we originally did comprehensive ablation studies, but they were omitted due to the space limit. We thought reporting more results on more tasks would be more important than reporting ablation studies. Apparently, we were wrong. We will add them back in the updated version, which will have 1 more page. Those ablation studies were systematically conducted on the LCQMC dataset (a large-scale Chinese question matching corpus).

8 **1.1 Training Strategies** In our proposed training strategy (BERT-glyph-joint), we first only fine-tune the BERT model using task-specific supervising signals. Next, we freeze BERT and then update parameters of the Glyph layer. Finally, we relax BERT and fine-tune the two models jointly. Baseline training strategies include (1) *Glyph-Joint*, in which BERT is not fine-tuned at the beginning, i.e., we first freeze BERT to train the glyph layer, and then jointly train both layers until convergence; and (2) the *joint* strategy, in which we directly train the two models together until convergence. Results are shown in Table 1. The proposed training strategy introduces a performance boost of F1 about +1.0 over the others.

15 **1.2 image-classification training objective** Table 2 explores the influence of the image-classification training objective. As can be seen, this auxiliary training objective introduces a +0.8 F1 performance boost.

17 **1.3 Structures of the task-specific output layer** We change transformers in the task-specific output layer to other structures such as BiLSTMs and CNNs to explore their effects. Results for different models on different tasks are shown in Table 3.

20 **1.4 CNN structures** Results for different CNN structures are shown in Table 4. As can be seen, the adoption of tianzige-CNN structure introduces a performance boost of F1 about +1.0.

| Strategy | Precision | Recall | F1 |
|---|---|---|---|
| BERT-glyph-joint | 86.8 | 91.2 | 88.8 |
| Glyph-Joint | 82.5 | 94.0 | 87.9 |
| joint | 81.5 | 95.1 | 87.8 |

Table 1: Impact of different training strategies.

| Strategy | Precision | Recall | F1 |
|---|---|---|---|
| W image-cls | 86.8 | 91.2 | 88.8 |
| WO image-cls | 83.9 | 93.6 | 88.4 |

Table 2: Impact of the image classification objective.

| Strategy | Precision | Recall | F1 |
|---|---|---|---|
| Transformers | 86.8 | 91.2 | 88.8 |
| BiLSMTs | 81.8 | 94.9 | 87.9 |
| CNNs | 81.5 | 94.8 | 87.6 |
| BiMPM | 81.1 | 94.6 | 87.3 |

Table 3: Impact of different structures for the task-specific output layer.

| | Precision | Recall | F1 |
|---|---|---|---|
| Tianzige-CNN | 86.8 | 91.2 | 88.8 |
| Kim 2014 | 85.7 | 90.4 | 87.9 |
| Vanilla-CNN | 85.3 | 89.8 | 87.4 |
| ResNet | 84.5 | 90.8 | 87.5 |

Table 4: Impact of different CNN structures.

## 2.1 First Reviewer

**2.1.1** Task details: We appreciate the helpful suggestions. To demonstrate the generalization power of the GLYCE model, we extensively tested our model on a wide range of NLP tasks. Experiments were conducted on 21 datasets across 7 different tasks. We will add all the details of each task in the appendix of the final version.

**2.1.2** Training details: we are sorry for the missing training details. Please refer to Section 1.1. We will add these details in the final version.

**2.1.3** More details about the glyph CNN itself: sorry for the confusion. The glyph-CNN is detailed in Section 2.2 in the original paper, but we will make it clearer in the updated version.

## 2.2 Second Reviewer

**2.2.1** Appropriateness: Generally, we think that Glyce is a perfect fit for NeurIPS. NeurIPS/NIPS has a long-standing reputation for presenting fundamental deep learning technology or methodology that improved a wide range of NLP tasks, e.g., *Sutskever et al., Sequence to Sequence Learning with Neural Networks, NIPS2014*; *Mikolov et al., Distributed Representations of Words and Phrases and their Compositionality, NIPS2013*. GLYCE is actually along this line of research. It offers a universal methodology to deal with character graph of logographic languages, and achieves SOTA results on 21 datasets across 7 tasks.

**2.2.2** Why visual features would help in certain cases: Sorry for the confusion. In logographic languages, the glyph of a character encodes semantic information. The meaning of a character can not only be inferred by its context (external), but also by its own glyph (internal). Glyph information is particularly helpful to model the meaning of rare characters, since there is not much context available to infer their meanings. For example, "鸣"(chirp) is composed of "口"(mouth) and "鸟"(bird), and "淼"(flood) is composed of three "水" (water). We can see that the glyph of a Chinese character is closely related to its meaning.

## 2.3 Third Reviewer

**2.3.1** details of transformers: thank you for the advice. We will include those details in the updated version.

**2.3.2** how many scripts in Table 1 are used: sorry for the confusion. We find that using all (i.e., 8) historical scripts is beneficial to all tasks, and thus we use all of them across all tasks.

**2.3.3** whether the original BERT is fine-tuned: sorry for the confusion. Please refer to Section 1.1 on this issue. We will add it in the updated version.

[Meta-Review · NeurIPS 2019]

We are accepting the paper but ask the authors to carefully address the reviewers's comments and revise the paper. Please try to improve the clarity of the presentation (experimental setup, training procedure, etc). Please also include qualitative evidence that the visual model is helpful for Chinese character embedding in addition to the quantitative results. This needs to go beyond the analysis and discussion in the rebuttal.